# Identification of a Sesquiterpene Lactone from *Arctium lappa* Leaves with Antioxidant Activity in Primary Human Muscle Cells

**DOI:** 10.3390/molecules26051328

**Published:** 2021-03-02

**Authors:** Nour El Khatib, Sylvie Morel, Gérald Hugon, Sylvie Rapior, Gilles Carnac, Nathalie Saint

**Affiliations:** 1PhyMedExp, University of Montpellier, CNRS, INSERM, F34295 Montpellier, France; elkhatib.nour@hotmail.com (N.E.K.); gerald.hugon@inserm.fr (G.H.); gilles.carnac@inserm.fr (G.C.); 2Laboratoire de Botanique, Phytochimie et Mycologie, CEFE, University of Montpellier, CNRS, EPHE, IRD, University of Paul Valéry Montpellier 3, F34093 Montpellier, France; sylvie.morel@umontpellier.fr (S.M.); sylvie.rapior@umontpellier.fr (S.R.)

**Keywords:** *Arctium lappa* L., burdock leaves, antioxidant, onopordopicrin, human myoblasts

## Abstract

Many pathologies affecting muscles (muscular dystrophies, sarcopenia, cachexia, renal insufficiency, obesity, diabetes type 2, etc.) are now clearly linked to mechanisms involving oxidative stress. In this context, there is a growing interest in exploring plants to find new natural antioxidants to prevent the appearance and the development of these muscle disorders. In this study, we investigated the antioxidant properties of *Arctium lappa* leaves in a model of primary human muscle cells exposed to H_2_O_2_ oxidative stress. We identified using bioassay-guided purification, onopordopicrin, a sesquiterpene lactone as the main molecule responsible for the antioxidant activity of *A. lappa* leaf extract. According to our findings, onopordopicrin inhibited the H_2_O_2_-mediated loss of muscle cell viability, by limiting the production of free radicals and abolishing DNA cellular damages. Moreover, we showed that onopordopicrin promoted the expression of the nuclear factor-erythroid-2-related factor 2 (Nrf2) downstream target protein heme oxygenase-1 (HO-1) in muscle cells. By using siRNA, we demonstrated that the inhibition of the expression of Nrf2 reduced the protective effect of onopordopicrin, indicating that the activation of the Nrf2/HO-1 signaling pathway mediates the antioxidant effect of onopordopicrin in primary human muscle cells. Therefore, our results suggest that onopordopicrin may be a potential therapeutic molecule to fight against oxidative stress in pathological specific muscle disorders.

## 1. Introduction

Oxidative stress, defined as an imbalance between pro-oxidants (reactive oxygen species—ROS) and antioxidants in favor of pro-oxidants is now proposed to be a common denominator in the pathogenesis of most chronic diseases such as cancer, coronary heart disease, neurodegenerative diseases, obesity, diabetes type 2, hypertension [1]. ROS, that are by-products of oxidative phosphorylation, would cause progressive damage to key cell macromolecules (lipids, proteins, and DNA), altering both their structure and function. In muscle-specific disorders such as muscular dystrophies and during aging, oxidative stress is proposed to be a major causal factor of muscle wasting [2,3,4,5,6]. The process of protein synthesis and degradation that controls muscle mass has been shown to be redox-sensitive [5]. Elevated oxidative stress can result in irreversible cellular damage. It has been associated with telomere shortening, the alteration of muscle cell proliferation, differentiation and muscle stem cell depletion during aging in a mice model of muscle degenerative disease [7,8,9,10]. Many mechanisms have been proposed to explain muscle degeneration caused by the accumulation of ROS. Among these, the loss of calcium homeostasis as a result of alteration of the ion channel membrane proteins, mitochondrial dysfunction, the activation of NF-κB which results in the expression of inflammatory genes, autophagy, a mechanism that allows the removal of defective proteins and organelles [6].

Despite the clinical relevance of antioxidant use as a therapeutic approach to improve muscle function and the great interest of a broad segment of the population in antioxidant supplementation, evidence in this field is very limited and the effect of antioxidants in delaying, preventing, or reversing the loss of muscle mass is unclear [11]. The difficulty is that the choice of antioxidant molecules is often based on their availability and not on their effectiveness. Generally, clinical trials are not preceded by pre-tests evaluating the effectiveness and toxicity of antioxidant molecules, tests that would allow selection of those that are most effective and the least toxic. In terms of toxicity, some antioxidants have been shown to have deleterious effects on the differentiation of cell precursors of skeletal, cardiac, smooth muscle and adipose tissues [12,13,14,15,16,17].

*Arctium lappa* L. (Asteraceae), commonly called burdock, is a popular plant in folk medicine widely used in China and other parts of the world. *Arctium lappa* (*A. lappa*) has recognized healthy effects and is applied to treat various diseases including diabetes, gout, rheumatism and skin problems [18,19,20,21]. Research on *A. lappa* extracts obtained from roots, seeds and leaves has identified several classes of bioactive secondary metabolites including lignans, flavonoids, quinic acids, phenolics, polyacetylenes, terpenoids, lactones and polysaccharides [22]. Many of the biological properties attributed to *A. lappa*, including antimutagenicity, anticancer, and antiaging, may originate from the ability of its components to reduce inflammation and oxidative stress [23,24,25]. As *A. lappa* leaves are poorly characterized compared to the other parts of the plants (roots, seeds and fruits) and few data are reported in literature, we decided to focus our research on leaf extracts of *A. lappa*. This choice was also supported by the fact that *A. lappa* leaves, being discarded following roots and seeds harvesting, become a plant waste material available for the valorization of bioactive compounds. In a previous study on *A. lappa* leaves, Franco et al. [26] showed that ethanol extracts present high antioxidant capacities in in vitro assays such as the scavenging of free radicals (DPPH assay) and oxygen radical absorbance capacity (ORAC assay). Another work also using the DPPH assay pointed out the hydroalcoholic extracts of *A. lappa* leaves as antioxidant [27]. In this study, Lou and collaborators showed that the antioxidant capacity was associated with the total phenolic content of extracts. Moreover, they suggested that compounds such as chlorogenic acid, o-hydroxybenzoic acid, caffeic acid, *p*-coumaric acid and rutin present in *A. lappa* leaves extracts could be involved in antioxidant activity. More recently, Kim et al. [28] showed that an *A. lappa* leaves extract has a protective effect on an age-related macular degeneration model and they suggested that this effect could be due to antioxidant properties of phenolic and flavonoid content present in the leaves extract. Thus, all these studies have reported a potential antioxidant ability of *A. lappa* leaves but no specific compound has been isolated from the extracts.

In this study, we first developed a cellular assay for testing antioxidant activities of *A. lappa* leaves in a model of stressed-human skeletal muscle cells with hydrogen peroxide (H_2_O_2)_, a strong pro-oxidant molecule. Then, we used a bioassay-guided fractionation to identify bioactive molecules from leaves of *A. lappa*. Finally, we studied the capacity of the identified antioxidant molecule to protect primary human muscle cells from oxidative stress, by measuring cell viability, levels of ROS, and DNA damage as well as the activation of the Nrf2 signaling pathway.

## 2. Results

### 2.1. Antioxidant Activity of Arctium lappa Leaf Extract and Bioactivity-Guided Separation of Antioxidant Compounds

We previously showed that the pro-oxidant molecule H_2_O_2_ increases the percentage of apoptotic cells in adherent cultures of primary human myoblasts (skeletal muscle precursor cells) [29]. We used this model of H_2_O_2_-treated human skeletal muscle cells to screen the antioxidant activity of an ethanolic *A. lappa* leaves extract (burdock leaves extract, BL). We pre-incubated human myoblasts with increasing concentrations of BL extract prior to the addition of a lethal concentration of H_2_O_2_ in the culture medium. BL pre-incubated at 70 and 140 µg/mL for 24 h with myoblasts drastically reducing cell death (Figure 1A). We determined that both these concentrations of BL were also able to efficiently decrease the levels of ROS produced by the cells following the addition of H_2_O_2_ (Figure 1B).

A bioassay-guided fractionation method combining a precipitation step with MeOH and CH_2_Cl_2_ as solvents and chromatography steps using LH 20 Sephadex and silica gels (see Materials and Methods 3.5 and Appendix A) was used to identify the compound(s) responsible for the antioxidant activity of the *A. lappa* leaves extract. After four steps of purification a pure antioxidant compound was obtained (Appendix A and Figure 2). The structure was elucidated by NMR (Appendix A) as onopordopicrin, previously described in *A. lappa* [30,31,32]) and other Asteraceae [33,34,35].

### 2.2. Antioxidant Capacity of the Sesquiterpene Lactone Onopordopicrin

#### 2.2.1. Onopordopicrin Protects Human Myoblasts against H_2_O_2_-Induced Stress

Before assessing onopordopicrin antioxidant activity, increasing concentrations of this molecule were incubated for 24 h with myoblasts to evaluate its cytotoxicity. We first measured the viability of cells (Figure 3A) and the accumulation of ROS (Figure 3B), highly reactive molecules associated with apoptotic cells. Based on both graphics (Figure 3A,B), we observed that onopordopicrin was toxic starting from 1 µg/mL with a decrease in cell viability and an increase in ROS quantity. Therefore, to avoid cytotoxicity we chose to incubate myoblasts with 0.125 µg/mL, 0.25 µg/mL and 0.5 µg/mL of onopordopicrin to perform the antioxidant assay on myoblasts. Data on Figure 3C indicate that pretreatment with onopordopicrin, concentrations ranging from 0.125 to 0.5 µg/mL, inhibited the H_2_O_2_-mediated loss of myoblast cell viability. We further determined whether onopordopicrin can abolish H_2_O_2_-induced DNA damage by analyzing the phosphorylation of γH2AX (*p*-γH2AX), a protein phosphorylated upon the DNA double-strand break. As shown in Figure 3D the protein expression of *p*-γH2AX in cells treated with H_2_O_2_ alone drastically increased. However, pretreating the cells with onopordopicrin at 0.25 µg/mL and 0.5 µg/mL reduced the H_2_O_2_-induced phosphorylation of γH2AX, indicating that onopordopicrin blocked DNA damages due to oxidative stress.

#### 2.2.2. Onopordopicrin Activates the Nrf2/HO-1 Signaling Pathway

To overcome oxidative stress, cells have several antioxidant defense mechanisms to maintain ROS at low physiological levels. One of them is the modulation of the nuclear factor erythroid-2-related factor 2 (Nrf2), a transcription factor that could mediate genes encoding drug transporters, antioxidants enzymes, anti-apoptotic proteins and detoxifying factors [36]. It is known that Nrf2 plays a protective role against oxidative stress through regulating anti-oxidative genes, such as heme oxygenase-1 (HO-1) [37]. Thus, we have evaluated whether the antioxidant activity of onopordopicrin was correlated with the activation of the Nrf2/HO-1 signaling pathway. By analyzing immunoblotting data, we found that the expression of HO-1 protein was upregulated in myoblasts treated with the onopordopicrin sample compared with untreated cells (Figure 4A). These results suggest that onopordopicrin can act as an activator of Nrf2 signaling pathway in myoblasts. Then, we investigated whether the Nrf2 signaling pathway was involved in the protection of onopordopicrin against H_2_O_2_-induced oxidative stress by using Nrf2 siRNA. We tested two Silencer Select siRNA of Nrf2, siNrf2-91 and siNrf2-93. siRNA 91 was the most effective in inhibiting Nrf2 (80% inhibition) and its target HO-1 (60% inhibition), and was therefore used for the following experiments (Figure 4B). We first tested the effect of dimethylfumarate (DMF), a well-known activator of Nrf2 pathway [38]. Figure 4C shows that DMF treatment protects myoblasts from H_2_O_2_-induced cell death. This protecting effect was lost in Nrf2 siRNA-transfected cells. Similarly, by pretreating myoblasts with onopordopicrin, we were able to protect cells with siRNA negative control (siCTRL) but not siNrf2 cells from H_2_O_2_-induced apoptosis (Figure 4D). In contrast, Tempol, a powerful synthetic antioxidant acting through the direct chelation of free radicals [39], reduced the number of dead cells both in siCTRL and in siNrf2 transfected cells treated with H_2_O_2_ (Figure 4D). These results demonstrate that onopordopicrin activates Nrf2/HO-1 signaling pathway in H_2_O_2_-treated myoblasts alleviating oxidative stress and improving oxidant resistance.

## 3. Discussion

In this study, we identified and characterized a molecule extracted from burdock leaves, onopordopicrin, as being a molecule with strong antioxidant activity in human muscle cells. *Arctium lappa*, a well-known traditional medicinal plant contains compounds that have shown many bioactive properties, such as anti-inflammation, antioxidant, anti-microbial, anti-viral and anticarcinogenic effects [22]. Previous studies, made with in vitro assays such as the DPPH assay, have provided evidence for the antioxidant activity of *A. lappa* leaves [26,27]. It has been suggested that the phenolic and flavonoid content present in the leaf extracts could be responsible for the antioxidant activity [27]. In our study, we used a bioassay-guided fractionation method to identify antioxidant compounds from an *A. lappa* leaves extract. Surprisingly, the main antioxidant able to protect the H_2_O_2_-treated myoblasts we identified was not a phenolic or flavonoid compound but a molecule called onopordopicrin belonging to the family of sesquiterpene lactones. Initially, onopordopicrin has been reported as a constituent in leaves of *Onopordum acanthium* [33]. This molecule has also been previously isolated from *A. lappa* leaves [30,31,32] and from aerial parts of *Arctium nemorosum* and *Onopordum illyricum* [34,35]. Sesquiterpene lactones are lipophilic secondary metabolites abundant in plants of the Asteraceae family. These molecules have been reported to possess a broad spectrum of biological activities including anti-inflammatory, phytotoxic, antibacterial, antiplasmodial, antioxidant and cytotoxic/anticancer properties [31,35,40,41,42,43,44,45,46]. In this study, we demonstrated that onopordopicrin was able to prevent myoblasts from oxidative stress leading to an increase in cell viability, decrease in ROS production and DNA cellular damages. Previously, we showed that taxodione, a diterpene extracted from stems of *Rosmarinus officinalis,* could protect human myoblasts against H_2_O_2_-induced oxidative stress damage [47]. We also demonstrated that taxodione is as effective as butylated hydroxytoluene (BHT), a synthetic antioxidant, in limiting lipid and protein oxidation in post mortem mouse and beef muscles [47]. We tested onopordopicrin in similar conditions to see if this molecule could reduce oxidation during the refrigerated storage of meat but no protection of lipids and proteins against oxidation has been observed (unpublished observations). These results suggest that onopordopicrin may not share the mechanisms of taxodione and BHT such as the direct chelation of free radicals to prevent muscle cells from oxidative stress. Onopordopicrin could be useful in the treatment of degenerative muscle diseases. Oxidative stress has been implicated in several skeletal muscle degenerative diseases, such as Duchenne muscular dystrophy (DMD), facioscapulohumeral dystrophy (FSHD), laminopathy and dystrophies related to mutation in collagen VI and dysferlin genes [6]. In addition, high oxidative stress could limit the effectiveness of therapeutic approaches in skeletal muscle [48,49,50,51,52]. The failure of cell therapy in initial clinical trials was mainly attributed to poor survival rates of transplanted myoblasts [53,54,55,56]. There is an interesting correlation between the redox status of myoblasts and their effectiveness in regenerating muscle tissue. Myoblasts are characterized by enhanced ROS levels, reduced levels of antioxidant and detoxifying enzymes and an increased sensitivity to oxidative stress [57]. The pre-treatment of myoblasts with antioxidant molecules improves graft survival and would be an interesting strategy to improve cell transplantation therapy ([49,50,51]).

Onopordopicrin isolated from *Onopordum illyricum* was shown to act as a potent inhibitor of the pro-inflammatory transcription factors NF-κB and STAT3. In addition, onopordopicrin was reported to enhance the activity of Nrf2, quantified using the HaCaT-ARE-Luc reporter gene suggesting a potential role of onopordopicrin on the expression of Nrf2 gene, the main regulator of the cellular antioxidant response [35]. Under stress conditions, Nrf2 uncouples from its endogenous inhibitor Kelch-like ECH-associated protein 1 (Keap1). Then, Nrf2 enters the nucleus and binds its cognate response element, the antioxidant response element (ARE), which upregulates the transcription of ARE–responsive genes, such as HO-1 or NQO1 (quinone oxidoreductase 1) having the ability to protect cells against oxidative stress. Thus, in this study, we investigated whether onopordopicrin could activate the endogenous Nrf2 signaling pathway in primary human muscle cells. By using siRNA, we demonstrated that the inhibition of Nrf2 reduces the expression level of Nrf2 downstream target protein HO-1 and abolishes the protective effect of onopordopicrin in muscle cells. These results indicate that the activation of the Nrf2/HO-1 signaling pathway mediates the antioxidant effect of onopordopicrin in primary human muscle cells. 

The activation of the Nrf2/HO-1 signaling pathway by onopordopicrin is a characteristic shared by other sesquiterpene lactones. For instance, the two sesquiterpene lactones hemistepsin A from *Hemistepta lyrata* and alantolactone from *Inula helenium* have been shown to activate the Nrf2/HO-1 pathway and to protect against oxidative stress in human keratinocytes and bronchial epithelial cells, respectively [45,58]. Another sesquiterpene lactone, parthenolide, found in feverfew (*Tanacetum parthenium*) and belonging, like onopordopicrin, to the germacrane subclass, was shown to have antioxidant properties by increasing the expressions levels of Nrf2 and HO-1 in H_2_O_2_-treated osteoblasts [59]. In a more recent study, it was demonstrated that parthenolide could also inhibit obesity and obesity-induced inflammatory responses in in vitro and in vivo models. For the in vivo study, the authors used a model of high-fat diet (HFD)-fed mice to show that the administration of parthenolide induced a reduction in body weight and white adipose tissues. Moreover, they demonstrated that the parthenolide-mediated anti-obese effect was connected to anti-inflammatory responses with the regulation of inflammatory cytokines. In addition. they showed that parthenolide suppressed HFD-mediated oxidative stress by regulating the level of antioxidant molecules via the activation of the Nrf2 signaling pathway [60]. Thus, it seems that an increasing number of recent studies demonstrated the activation of Nrf2 by sesquiterpene lactones like onopordopicrin but how these molecules can activate Nrf2 is still not well understood. It is believed that the methylene-ɣ-lactone group of sesquiterpene lactones is the main responsible for biological activities as anti-inflammatory [35,61], cytotoxic [34] or antitrypanosomal [34]. Indeed, the methylene-ɣ-lactone group can react as a thia-Michael acceptor. Moreover, the opening of the ɣ-lactone is associated with a decrease in Nrf2 activation [35]. The 2-(hydroxymethyl)acrylate side chain represents a second Michael acceptor site in the structure of onopordopicrin. This chain seems to be important for activities [34,61], but it appears to be less reactive than the methylene-ɣ-lactone group in NMR-based cysteamine assay [35]. For onopordopicrin, thia-addition is observed quickly and only on the methylene-ɣ-lactone group, whereas for carmanin, a sesquiterpene without lactone, the addition was made on the 2-(hydroxymethyl)acrylate side chain very slowly [35]. Moreover, costunolide, a sesquiterpene lactone without a side chain, was an effective Nrf2 activator in PC12 cells [62] and finally elevated the protein expression of NQO1 (quinone oxidoreductase 1), TrxR (thioredoxin reductase), Trx (thioredoxin) and HO-1. This suggests that a Michael addition reaction between the methylene-ɣ-lactone group in costunolide and the cysteine residue in Keap1 is responsible for the activation of Nrf2 [62]. The same mechanism can occur with onopordopicrin in muscle cells. Further experiments with a large panel of sesquiterpenes could confirm that the methylene-ɣ-lactone group is the major site for the activation of Nrf2. The role of the 2-(hydroxymethyl) acrylate side chain could be determined for the reaction with Keap1.

In addition to their antioxidant and anti-inflammatory activities, sesquiterpene lactones have been shown to have cytotoxic/anticancer effects. In this context, parthenolide is one of the most studied sesquiterpene lactones and is considered as a potential antitumoral agent [63]. Sesquiterpene lactones have been shown to inhibit proliferation, induce apoptosis, promote cell cycle arrest and inhibit metastasis [64]. These processes would result from the disruption of the cellular redox balance and induction of oxidative stress in cancer cells [63]. In our study we showed that onopordopicrin, at concentrations superior to 0.5 µg/mL, induced ROS accumulation and cell cytotoxicity (Figure 3A,B), demonstrating that onopordopicrin can act as a pro-oxidant under certain conditions at high concentrations. These results are in agreement with a previous study showing that onopodopicrin has antiproliferative activity in tumor cells [31]. Therefore, it would be interesting to evaluate in further studies whether onopordopicrin could induce oxidative stress in cancer cells as it is proposed for other sesquiterpene lactones like parthenolide.

## 4. Materials and Methods

### 4.1. General Experimental Procedure

Flash column chromatography was performed using a Spot Liquid Chromatography Flash Instrument (Armen Instrument, Saint-Avé, France) equipped with a quaternary pump, an UV/visible spectrophotometer and a fraction collector. The ^1^H-NMR, ^13^C-NMR, COSY, HSQC, and HMBC spectra were recorded on a BRUKER Avance III-600 MHz NMR spectrometer. The coupling constants (*J*) are estimated in Hertz.

### 4.2. Reagent and Standards

Cyclohexane (99.8%), chloroform (99%), dichloromethane (99.9%), deuterated methanol (99.8%), DMSO (99.9%) and Tempol were purchased from Sigma-Aldrich (Steinheim, Germany). Acetonitrile (99.9%) was purchased from Chromasolv (Seelze, Germany). Formic acid (98%), ethyl acetate (99%) and acetone (99.5%) were purchased from Panreac (Barcelona, Spain). Ethanol (99.9%) was purchased from VWR BDH Prolabo (Monroeville, PA, USA).

### 4.3. Plant Material

*Arctium lappa* was collected in a collective garden of Montpellier “Jardin Des Aubes” (France). The plants were identified by botanists. Air-dried leaves were ground to a homogeneous powder and directly extracted. A voucher specimen was deposited in the Laboratoire de Botanique, Phytochimie et Mycologie, Faculty of Pharmacy, Montpellier (France) as number B50-2018.

### 4.4. Extraction

Fifty grams (50 g) of air-dried leaves were macerated in the dark at room temperature with 420 mL of absolute ethanol with manual agitation every 24 h. After 7 days, the leaf extract was filtered. Evaporation under reduced pressure to dryness yielded 2.75 g of ethanolic extract, named BL (burdock leaves). The dry extract was kept at −20 °C until analysis and purification.

### 4.5. Bioassay-Guided Isolation of Onopordopicrin from the Burdock Leaves Extract

To identify the compound(s) responsible for the antioxidant activity of the A. lappa leaves extract (BL extract), we used a bioassay-guided fractionation method. At each step of the purification, we evaluated the ability of the obtained fractions to protect myoblasts against H_2_O_2_-induced cell death (assay described in Section 4.8). The BL extract (2.75 g), solubilized in CH_2_Cl_2_, was submitted to permeation gel chromatography on a Sephadex LH-20 column (2.3 × 45.5 cm, 50 g LH-20). The elution was performed from 100% CH_2_Cl_2_ to 100% methanol (MeOH) with increasing concentrations of MeOH as follow: 2.5% (100 mL), 5% (400 mL), 20% (30 mL), 50% (40 mL), 100% (40 mL). The elution was finalized with 100% of water (100 mL) then 100% acetone (50 mL). After thin-layer chromatography (TLC) analysis, twelve fractions were obtained (1 to 12) (diagram 1 in the Supplementary Data). We found that fraction 3 (47 mg) and fraction 7 (183 mg) were responsible for BL extract antioxidant activity by reducing H_2_O_2_-treated primary muscle cell death (Figure 2A). Fraction 7 being more important in terms of weight, we decided to follow the bioactivity-guided purification with this fraction. Fraction 7 was precipitated in MeOH and then in CH_2_Cl_2_ to afford three fractions (Appendix A). The CH_2_Cl_2_ soluble fraction 1 had the higher antioxidant activity among these three fractions (Figure 2B) and was further purified on a Sephadex LH-20 gel (column: 1.1 × 40 cm, 12 g LH-20). The elution was completed with a mixture of CH_2_Cl_2_/MeOH (100:0 to 90:10) and then 100% of acetonitrile, yielding seven fractions (1 to 7) (Appendix A). Fraction 2 (46 mg), the most bioactive fraction reducing muscle cell death to the level of controlled cells (Figure 2C), was finally purified on normal phase flash chromatography (Interchim PuriFlash SIHP 15 µm–4 g, flow rate 1 mL/min, 1 mL/fraction) eluted with a mixture of CH_2_Cl_2_/MeOH (100:0 to 70:30 in 1, 5 and 10% stepwise) to obtain an antioxidant fraction (fraction 4) (Figure 2D) containing 5 mg of pure compound. Analysis of 1H-, 13C-NMR and 2D-NMR data (Appendix A) allowed us to identify this bioactive molecule as onopordopicrin as previously described [22,23,24]. The relative purity of onopordopicrin at 254 nm is 95.9% and we calculated that the isolated molecule comprised 0.18% (5 mg) of BL (2.75 g).

### 4.6. High-Performance Liquid Chromatography (HPLC) Analysis

Chromatographic separation and detection were performed on an Ultimate 3000 (Thermo Fisher Scientific Inc., San Jose, CA, USA) instrument that included a quaternary pump, a degasser, an automatic sampler and an UV Diode Array Detector. The system was operated using the Chromleon software, version 7.0. Chromatographic separation was achieved on an ODS Hypersyl C18 column (250 mm × 4.6 mm, 5 μm, Thermo Fisher Scientific Inc., San Jose, CA, USA), with a column temperature maintained at 35 °C. Fractions were eluted at a flow rate of 1 mL/min, using solvent A (water/formic acid 99.9:0.1 *v*/*v*) and solvent B (methanol). The gradient used for the analysis of extracts and fractions was initial mobile phase 5% of B reaching 100% in 40 min, with a flow rate of 1 mL/min. The UV/vis spectra were recorded in the 200–400 nm range and chromatograms were acquired at 254, 280 and 330 nm. BL crude extract and fractions were analyzed at 5 mg/mL. Onopordopicrin was analyzed at 1 mg/mL and its purity was determined at 254 nm.

### 4.7. Primary Cultures of Human Myoblasts

The quadriceps muscle biopsy was from one healthy adult (AFM-BTR “Banque de tissus pour la recherche”). Muscle biopsies (50 mg) were scissor minced and tissue fragments were plated in collagen-coated dishes. Explants were anchored to the dish by a thin layer of Matrigel^®^ (Corning, New York, NY, USA) and maintained in growth medium composed of DMEM/F12 medium with 10% fetal bovine serum (FBS), 0.1% Ultroser G and 1 ng/mL of human basic fibroblast growth factor (proliferation medium) of at 37° C in 95% humidified air, with 5% CO2. After 6 to 8 days, cells migrated out of the explants. Migrating cells were enzymatically harvested using dispase (Corning, NY, USA) and subcultured in growth medium. To isolate the myoblasts, the cells were incubated with mouse monoclonal antibody against CD56 (ref: 559043, BD Biosciences, Le Pont-de-Claix, France) and then with anti-mouse microbeads (ref: 130.048.402, Miltenyi Biotec, Paris, France), and purified with magnetic activated cell sorter (MACS) technology according to the manufacturer’s instructions (Miltenyi Biotec, Paris, France). After purification, the myoblasts (CD56+) were at least enriched at 99%. Myoblasts were cultured on collagen 1 (rat tail tendon)-coated dishes (ref: 356456, Corning, NY, USA) in DMEM/F12 medium with 10% fetal bovine serum (FBS), 0.1% Ultroser G and 1 ng/mL of human basic fibroblast growth factor (proliferation medium), as previously described [65]. 

### 4.8. Cell Death and ROS Quantification

Myoblasts were seeded at 1.10^5^ in 35 mm collagen-coated dishes, cultured in proliferation medium for 24 h, pre-incubated or not with the tested compounds for 24 h and then incubated or not with a lethal concentration of hydrogen peroxide (H_2_O_2_), a strong pro-oxidant/pro-apoptotic compound, for 24 h. The optimal H_2_O_2_ concentration was the concentration required to kill between 30 and 50% of total cells and was established before each experiment. In general, myoblasts were incubated with 120 µM H_2_O_2_. Total cells (non-adherent cells and adherent cells) were labeled with the Muse^®^ Count and Viability Kit (ref: MCH100102, Luminex, Austin, TX, USA) to detect apoptotic cells, and ROS was quantified with the Muse^®^ Oxidative Stress Kit (ref: MCH 100111, Luminex, Austin, TX, USA), followed by analysis with a Fluorescence Activated Cell Sorting (FACS) Muse apparatus (Millipore, Molsheim, France).

### 4.9. siRNA Transfections

Silencer Select siRNA negative control (N°1) and Silencer Select siRNA Nrf2 (siNrf2) (s9491 and s9493) were purchased from ThermoFisher scientific (San Jose, CA, USA). Human myoblasts were seed in 35 mm dishes at 1.10^5^ cells and transfected 24 h later with a mix of Lipofectamine RNAiMAX reagent (6 µL) (Life Technologies, Saint Aubin, France) and 5 nM of siRNA (siCTRL or siNrf2-s9491) for 24 to 48 h.

### 4.10. Western Blotting

Myoblasts were seeded at 1.10^5^ in 35 mm collagen-coated dishes, cultured in proliferation medium for 24 h, pre-incubated or not with onopordopicrin for 24 h and then incubated or not with a sub-lethal concentration (70 μM) of hydrogen peroxide (H_2_O_2_) for 24 h. Protein extracts (25 µg/well) were separated by SDS-PAGE gel electrophoresis one hour at 25 mA/gel using Mini Protean Precasts Gel 4–15% (Bio-Rad) and transferred in 5 min (1.3 A, 25 V) with Transblot turbo transfert system (Bio-Rad, Schiltigheim, France), to nitrocellulose membranes (Transblot turbo transfert pack 0.2 µm nitrocellulose, Bio-Rad, Schiltigheim, France), blocked at room temperature with Odyssey blocking buffer (Eurobio Scientific, Les Ulis, France) and probed with the rabbit polyclonal anti-heme oxygenase 1 (H-O1) (ref: 43966S; 1/2000; Cell Signalling, Danvers, MA, USA) antibodies, rabbit polyclonal anti phosphorylated γ-H2AX (ref: 9718S; 1/3000; Cell Signalling, Danvers, MA, USA) antibodies and rabbit polyclonal anti Nrf2 (ref: 127215; 1/1000; Cell Signalling, Danvers, MA, USA) antibodies followed by IRDye^®^ 680RD and IRDye^®^ 800RD secondary antibodies (Eurobio Scientific, Les Ulis, France). Fluorescence was quantified with the Odyssey software. Data were normalized to α tubulin expression (alpha tubulin mouse antibody (ref: T9026; 1/10,000; Sigma-Aldrich, Lyon, France).

### 4.11. Statistical Analysis

Statistical analysis was done with the GraphPad Prism 6.0 software (GraphPad Software Inc., San Diego, CA, USA). All experiments were performed in triplicate. Error bars represent the SD of the mean. Statistical significance was determined using one-way ANOVA; *p* < 0.05 (*), *p* < 0.01 (**), *p* < 0.001 (***) and *p* < 0.0001 (****) were considered significant.

## 5. Conclusions

By using a bioactivity-guided fractionation, we identified onopordopicrin as the main molecule of the leaves extract of *A. lappa* having antioxidant ability to protect muscle cells from H_2_O_2_-induced stress, through the activation of the Nrf2/HO-1 signaling pathway. In the future, the role of onopordopicrin in protecting skeletal muscle cells from stress-induced disorders could be evaluated in ex vivo models, such as human muscle cells derived from myopathic patients presenting an exacerbated susceptibility to oxidative stress and in vivo, on animal models of human muscle pathologies.

## Figures and Tables

**Figure 1 molecules-26-01328-f001:**
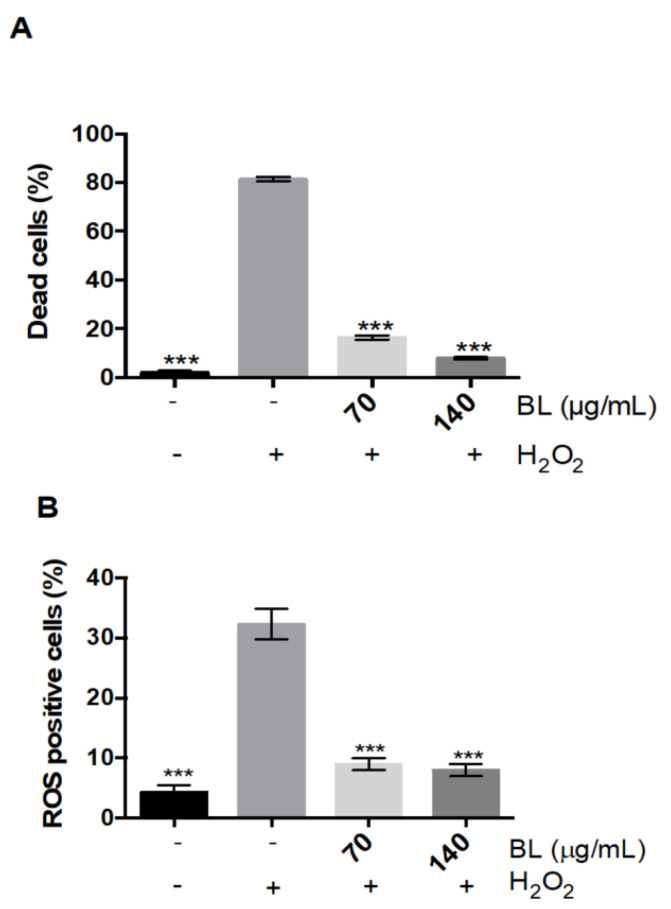
*A. lappa* leaves extract protects human myoblasts from induced oxidative stress. Cell death (**A**) and ROS (**B**) quantification (percentage of all cells) in human myoblasts that were incubated with the *A. lappa* leaves extract (BL) at indicated concentrations prior to incubation with H_2_O_2_. All data represent the means ± SD of three independent experiments; *p* < 0.001 (***) compared with H_2_O_2_ (one-way ANOVA).

**Figure 2 molecules-26-01328-f002:**
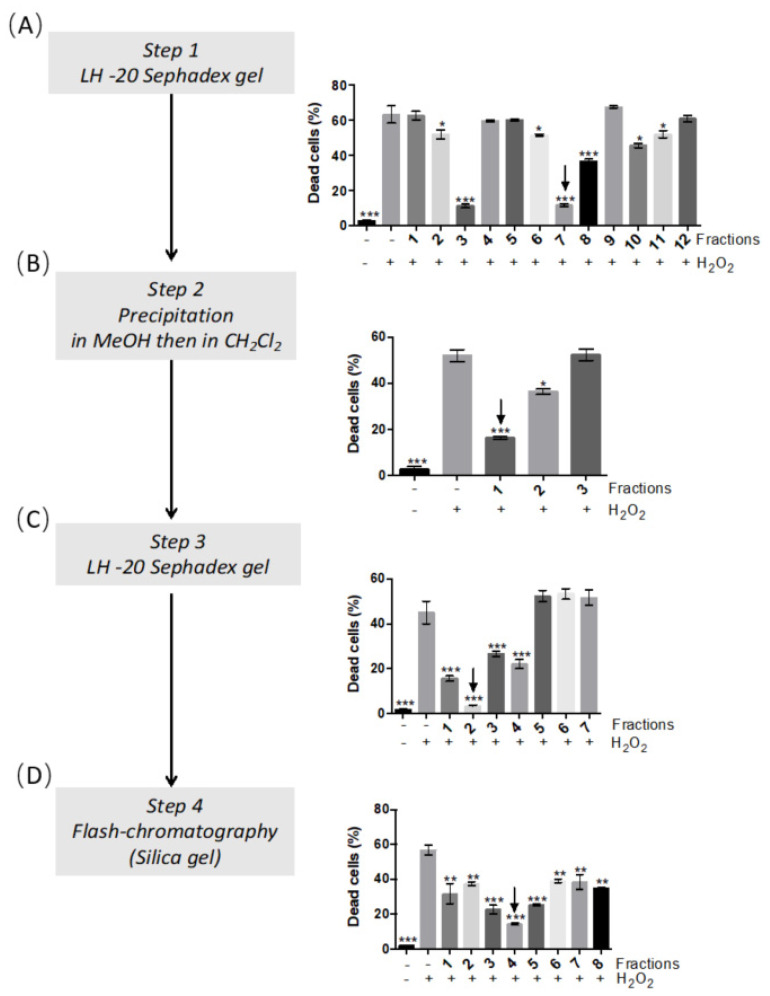
Bioassay-guided isolation of the antioxidant compound from the *A. lappa* leaves extract (BL extract). On the left: each step of the BL extract purification; on the right: the associated bioassay on myoblasts with the fractions obtained from the different steps of purification. Cell death quantification (percentage of all cells) in human myoblasts incubated with BL extract fractions at 1 μg/mL (step 1 (**A**) and 2 (**B**)) or 0.5 μg/mL (step 3 (**C**) and 4 (**D**)) prior to incubation with H_2_O_2_ All data represent the means ± SD of three independent experiments; *p* < 0.05 (*) *p* < 0.01 (**) *p* < 0.001 (***) compared with H_2_O_2_ (one-way ANOVA).

**Figure 3 molecules-26-01328-f003:**
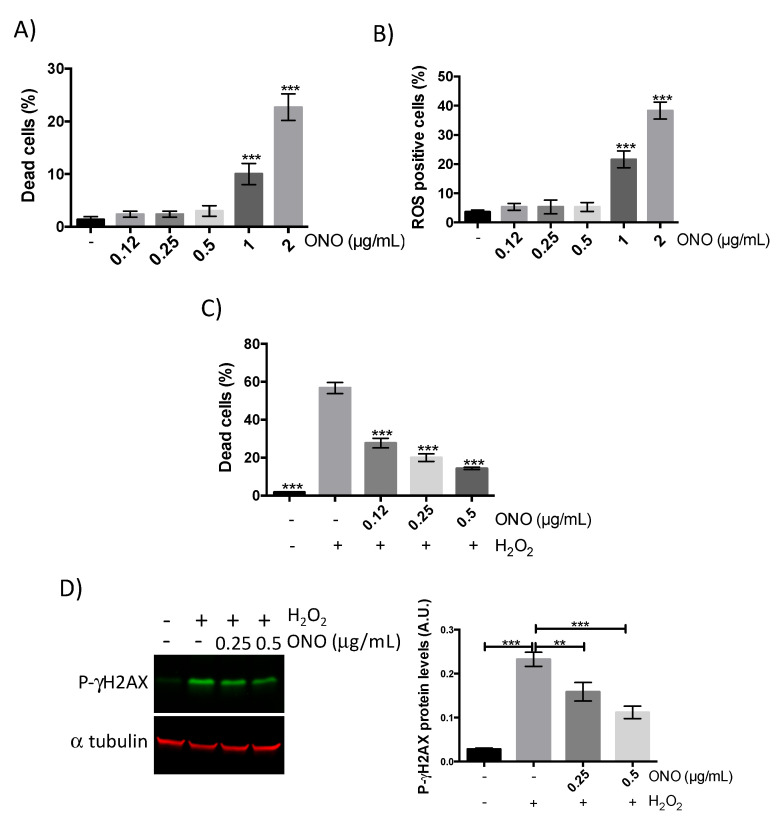
Onopordopicrin has a strong antioxidant activity on human muscle cells. Cell death (**A**,**C**) and reactive oxygen species (ROS) (**B**) quantification (percentage of all cells) in human myoblasts treated with onopordopicrin (ONO) at indicated concentrations prior to incubation with lethal concentration of H_2_O_2_; (**D**) human myoblasts treated with onopordopicrin (ONO) at indicated concentrations prior to incubation with sub-lethal concentration of H_2_O_2_. On the left, Western blot analysis of phosphorylated γH2AX protein level; on the right, quantification of the Western blot data, A.U. for arbitrary unit represents the ratio of *p*-γH2AX protein level on α tubulin protein level. All data represent the means ± SD of three independent experiments; *p* < 0.01 (**) *p* < 0.001 (***) compared with H_2_O_2_ in A, B and C (one-way ANOVA).

**Figure 4 molecules-26-01328-f004:**
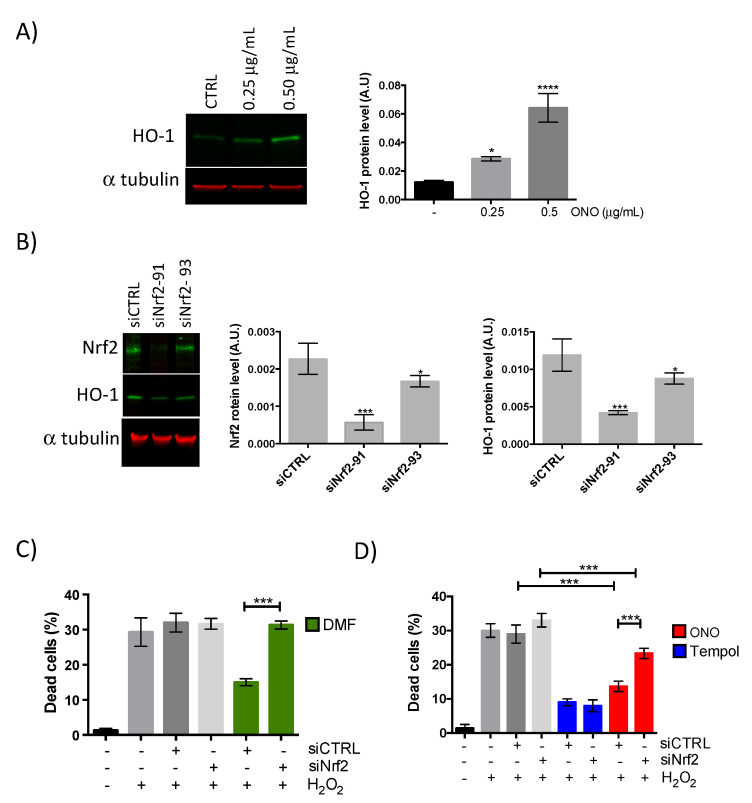
Onopordopicrin activates the Nrf2/HO-1 signaling pathway. (**A**) On the left, Western blot analysis of HO-1 protein level in human myoblast treated with onopordopicrin during 24 h at the indicated concentrations; CTRL: cells not incubated with onopordopicrin; on the right, quantification of Western blot data, A.U. for arbitrary unit represents the ratio of HO-1 protein level on α tubulin protein level. (**B**) On the left, Western blot analysis of HO-1 and Nrf2 levels in human myoblasts transfected with siNrf2-91 and siNrf2-93; on the right, quantification of the Western blot data, A.U. for arbitrary unit represents the ratio of Nrf2 and HO-1 protein levels on α tubulin protein level. (**C**,**D**). Cell death quantification (percentage of all cells) in human myoblasts transfected with Nrf2 siRNA or with siRNA negative control (siCTRL) and treated with dimethylfumarate (DMF) (20 µM) or Tempol (50 µM) or onopordopicrin (0.5 µg/mL) prior to incubation with H_2_O_2_. All data represent the means ± SD of three independent experiments; *p* < 0.05 (*) *p* < 0.001 (***) *p* < 0.0001 (****) compared to untreated cells in A and compared to siCTRL in B (one-way ANOVA).

## Data Availability

All the data are included on this paper.

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
