# Peer review of "Identification of a Sesquiterpene Lactone from Arctium lappa Leaves with Antioxidant Activity in Primary Human Muscle Cells"

_molecules, 2021, doi:10.3390/molecules26051328_

Round 1

Reviewer 1 Report

The authors describe onopordopicrin, as extract from Arctium lappa leaves, with antioxidant activity in primary human muscle cells. Antioxidant activity is provided by the ability of onopordopicrin to inhibit the H2O2 mediated loss of muscle cell viability, and the increased expression of Nrf2 downstream target protein HO-1.

The reviewer has minor comments:

  1. Line 37 and following: As the authors mention several cell-signaling dependent mechanisms here, it would be nice to have some of the responsible target proteins of the cell signal pathway mentioned here.
  2. Line 52: As the reference which is used to support all these “healthy effects” is already 11 years old, it would be better to have some follow up studies cited here additionally, to show recent knowledge in this field.
  3. Line 57: To convince the reader from real effects of the biological properties, it needs some more convincing references here, concerning anti-cancer, antimutagenicity, especially more actual references as this cited review was published 20 years ago.
  4. Chapter Results and Discussion: Please be consistent when using “burdock leaf” (BL), or “Arcitium lappa leaf”. Within the text the authors use BL and switch to Arcitium lappa in the figure legend of figure 1. The figure legend is confusing, as it is not clear which concentration is the BL concentration and which concentration was used for H2O2. The reviewer would recommend to make use of the color legend for example, by indicating the BL concentration by color and not by x-axis.
  5. Figures general: The Reviewer would recommend to use the SD error bars in both directions. Please be consistent with the style of error bars and connecting lines. Some appear thicker, then others.
  6. Table 1: In the Reviewers eyes, this table is not relevant and could, if possible, be put in a supplementary file.
  7. Line 147: The authors mention “ONO” here for the first time, without explaining what it means and although the authors had the chance several times on the previous pages of the manuscript to introduce this abbreviation, and even more important: The whole manuscript needs revision in terms of consistent wording (line152: onopordopicrin, line 156: ONO, line 157: onopordopicrin)
  8. Figure3: There is something wrong with the y-axis labeling.
  9. Figure 4a: Why did the authors show the loading control (revert) here, and not in figure 3? Please be consistent in the figure style and results presentation. The appearance of the graph in figure 4a seems to be somehow compressed, as the labeling of the y-axis appears different from the rest. Please explain A.U. to the reader in the figure legends containing western blots. Please use the same y-axis range within one figure for better comparability (death cells % B=40, C=30). In the reviewer’s eyes, four stars as significance indicator appears uncommon, as three (p=0.001) is by far enough. By the way, this is what the authors indicate in the figure legend. Please be consistent and check the whole manuscript for careless mistakes.

The reviewer has major comments:

  1. Results and Discussion general: It is in the reviewers eyes very uncommon to use results and discussion at the same chapter level, as this appears very disorganized to the reader. In the Reviewers eyes, and the common practice, results should clarify the facts without further references and the discussion should bring this in context with other working groups and literature. This is a general point the editor has to clarify for his journal.
  2. Materials and methods general: Also the materials and methods part seems to be mixed up with the results and discussion part: line 100-103 belongs the methods part in the reviewers eyes. The editor has to decide about this structural issue, and the whole manuscript has to be rearranged in the reviewers eyes, concerning the results, discussion and materials methods part.
  3. Line 107 – 117 and figure2: The explanations for the different methods and the different fractions belongs to the methods part. A pictogram for the general workflow and the fractions which were obtained and used for the further analysis would be very helpful. The label of the BL7 sample and the following BL7.1.1 and so on is totally confusing and needs revision to clarify what was done here. The arrow seems ok for the reviewer to indicate what is important here, but the labeling on the x-axis is not ok. Furthermore, the indication of 120µm of H2O2 as lethal concentration is redundant and should be explained only once in the methods section, the same for controls.
  4. Line 147: The authors mention “ONO” here for the first time, without explaining what it means and although the authors had the chance several times on the previous pages of the manuscript to in in the manuscript.
  5. General: The reviewer can´t find an explanation about the number of experiments which were carried out, so there is no “n” given. This is important to judge the relevance of the experiments and the authors have to explain, why they performed the experiments concerning the p-ƳH2AX Westernblot only twice. This is an important finding which is only weakly supported by an “n=2” and no relevance could be attributed to this finding. The authors should add a limitations section.
  6. Line 191: The authors state a correlation here, but no correlation was calculated. Thus, please show the significant or not significant correlation, or just say something like “associated with”, but prevent the term correlation if no statistics are available.
  7. Line 199: Why do the authors not show this very important result? An 80% reduction has to be shown, as it is one of the strongest effects the authors reporting here. Thus “data not shown” is not an option. Please show these western blot results and clarify how 80% reduction was calculated.
  8. The reviewer is missing more literature and clarification about the available results in animals. It appears like this whole topic has never been investigated in vivo. In line 229 there is a hind to an in vivo study, but no details about species and the detailed outcome. In the reviewer’s eyes, this manuscript needs extensive editing in terms of relevance for the clinics. If there is no evidence out there so far concerning the way of action in vivo, it is hard to figure out the relevance of this antioxidant.
  9. The reviewer is missing a limitations section.

Author Response

Reviewer 1

We would like to thank the reviewer for its constructive criticisms and suggestions, and we are confident that we have now addressed to the majority of its concerns.

The reviewer has minor comments:

  1. Line 37 and following: As the authors mention several cell-signaling dependent mechanisms here, it would be nice to have some of the responsible target proteins of the cell signal pathway mentioned here.

Response 1: We added sentences with the target proteins of the cell signal pathway we mentioned in the text.

  1. Line 52: As the reference which is used to support all these “healthy effects” is already 11 years old, it would be better to have some follow up studies cited here additionally, to show recent knowledge in this field.

Response 2: We added more recent references.

  1. Line 57: To convince the reader from real effects of the biological properties, it needs some more convincing references here, concerning anti-cancer, antimutagenicity, especially more actual references as this cited review was published 20 years ago.

Response 3: We added more convincing and more recent references.

  1. Chapter Results and Discussion: Please be consistent when using “burdock leaf” (BL), or “Arcitium lappa leaf”. Within the text the authors use BL and switch to Arcitium lappa in the figure legend of figure 1. The figure legend is confusing, as it is not clear which concentration is the BL concentration and which concentration was used for H2O2. The reviewer would recommend to make use of the color legend for example, by indicating the BL concentration by color and not by x-axis.

Response 4: We changed Arctium lappa leaf with burdock leaf in the legend of figure 1. We changed also the figures and the figure legends of the manuscript with color legend and + and – signs for better comprehension.

  1. Figures general: The Reviewer would recommend to use the SD error bars in both directions. Please be consistent with the style of error bars and connecting lines. Some appear thicker, then others.

Response 5 : We changed the errors bars and used the SD errors bars in both directions. We checked errors bars and connecting lines thickness.

  1. Table 1: In the Reviewers eyes, this table is not relevant and could, if possible, be put in a supplementary file.

Response 6: We put the table 1 in supplementary data.

  1. Line 147: The authors mention “ONO” here for the first time, without explaining what it means and although the authors had the chance several times on the previous pages of the manuscript to introduce this abbreviation, and even more important: The whole manuscript needs revision in terms of consistent wording (line152: onopordopicrin, line 156: ONO, line 157: onopordopicrin)

Response 7: We made the choice to write ‘onopordopicrin’ in all the text, except in the legends of the figures we kept the abbreviation ONO.

  1. Figure3: There is something wrong with the y-axis labeling.

Response 8: We made corrections in the y-axis labeling of figure 3.

  1. Figure 4a: Why did the authors show the loading control (revert) here, and not in figure 3? Please be consistent in the figure style and results presentation. The appearance of the graph in figure 4a seems to be somehow compressed, as the labeling of the y-axis appears different from the rest. Please explain A.U. to the reader in the figure legends containing western blots. Please use the same y-axis range within one figure for better comparability (death cells % B=40, C=30). In the reviewer’s eyes, four stars as significance indicator appears uncommon, as three (p=0.001) is by far enough. By the way, this is what the authors indicate in the figure legend. Please be consistent and check the whole manuscript for careless mistakes.

Response 9: We made a new figure 4. We made new western blot with alpha tubulin instead of the revert as loading control. We explained A.U. in the legend of the western blot. We used the same y-axis range within the figure 4 and we added four stars as significant indicator in the legend of figure 4.

 The reviewer has major comments:

  1. Results and Discussion general: It is in the reviewers eyes very uncommon to use results and discussion at the same chapter level, as this appears very disorganized to the reader. In the Reviewers eyes, and the common practice, results should clarify the facts without further references and the discussion should bring this in context with other working groups and literature. This is a general point the editor has to clarify for his journal.

Response 1: We changed the text and wrote the results and discussion in different chapters.

  1. Materials and methods general: Also the materials and methods part seems to be mixed up with the results and discussion part: line 100-103 belongs the methods part in the reviewers eyes. The editor has to decide about this structural issue, and the whole manuscript has to be rearranged in the reviewers eyes, concerning the results, discussion and materials methods part.
  2. Line 107 – 117 and figure2: The explanations for the different methods and the different fractions belongs to the methods part. A pictogram for the general workflow and the fractions which were obtained and used for the further analysis would be very helpful. The label of the BL7 sample and the following BL7.1.1 and so on is totally confusing and needs revision to clarify what was done here. The arrow seems ok for the reviewer to indicate what is important here, but the labeling on the x-axis is not ok. Furthermore, the indication of 120µm of H2O2 as lethal concentration is redundant and should be explained only once in the methods section, the same for controls.

Response 2 and 3: The text describing the fractionation of burdock leaves extract in ‘results’ section has been moved to ‘Materials and Methods’.  We made a diagram to describe the different steps of the bio-guided fractionation; this scheme is presented in supplementary data. We replaced the labels BL7 and the following BL7.1.1 by the number of the fractions we tested, in agreement with the scheme of the fractionation presented in supplementary data. We added the indication of 120 µM of H2O2 in the ‘Materials and Methods’ part and removed it from the legends of the figures.

  1. Line 147: The authors mention “ONO” here for the first time, without explaining what it means and although the authors had the chance several times on the previous pages of the manuscript to in in the manuscript.

Response 4: We wrote ‘onopodorpicrin’ in all the text. We kept the abbreviation ONO only in the figures and in the figure’s legends.

  1. General: The reviewer can´t find an explanation about the number of experiments which were carried out, so there is no “n” given. This is important to judge the relevance of the experiments and the authors have to explain, why they performed the experiments concerning the p-ƳH2AX Western blot only twice. This is an important finding which is only weakly supported by an “n=2” and no relevance could be attributed to this finding. The authors should add a limitations section.

Response 5 : We added the number of independent experiments that have been done in the legends of the figures. We performed new experiments of western blots concerning p-ƳH2AX. These new experiments are now included in a new figure 3D with three independent experiments.

  1. Line 191: The authors state a correlation here, but no correlation was calculated. Thus, please show the significant or not significant correlation, or just say something like “associated with”, but prevent the term correlation if no statistics are available.

Response 6: We replaced ‘correlated’ by ‘associated with’.

  1. Line 199: Why do the authors not show this very important result? An 80% reduction has to be shown, as it is one of the strongest effects the authors reporting here. Thus “data not shown” is not an option. Please show these western blot results and clarify how 80% reduction was calculated.

Response 7:  We add the results of Western Blot with two siRNAs (siNrf2-91 and siNrf2-93) in Figure 4B and we show also the quantification of this Western Blot indicating for siNrf2-91 that we have 80% of reduction in Nrf2 protein level and 60% of reduction of HO-1 protein level after normalization with alpha tubulin, our loading control.

  1. The reviewer is missing more literature and clarification about the available results in animals. It appears like this whole topic has never been investigated in vivo. In line 229 there is a hind to an in vivo study, but no details about species and the detailed outcome. In the reviewer’s eyes, this manuscript needs extensive editing in terms of relevance for the clinics. If there is no evidence out there so far concerning the way of action in vivo, it is hard to figure out the relevance of this antioxidant.

Response 8: We added more details for the cited reference (Kim et al 2019) that reported results on an in vivo model for treating inflammatory effects of obesity. This part of text is now in the discussion chapter.

  1. The reviewer is missing a limitations section.

Response 9: We wrote no limitations section, but we mentioned in the discussion chapter that ‘the antioxidant efficacy of onopordopicrin should be evaluated in vivo on animal models of skeletal muscle degenerative diseases and ex vivo on human cells derived from patients with myopathies which present an exacerbated susceptibility to oxidative stress’.

Reviewer 2 Report

The manuscript of El Katib et al present the identification of a biological compounds from Arctium lappa leaves and characterized the effects on cultured human muscle cells

Overall, the manuscript contains new interesting data, the studies are internally consistent and evidences are supported by clear experiments.

I have several concerns about the manuscript.

Major comments:

  • on all figures, labelling of the x axis is a bit confusing, some bars labelled with H2O2, but H2O2 also appears as an adding component for other bars. In figure 1 for example, concentration is indicated for 2 bars but not for the others. In figure 2, H2O2 line does not reach all fractions. Please modify figure presentation for clarify.
  • The muscle cells were used only as myoblasts, and the authors never studies differentiated myotubes (while differentiation is included in Mat&Meths). Proliferating myoblasts represent only a tiny proportion in muscle tissue, and exploration of differentiated myotubes could be of major interest. Please discuss the experimental choice.
  • On the human material, all experiments were performed on cells from a single donor. As individual specificity cannot be excluded, limitation of the study should be discussed.
  • The number of replicates for each experiment is difficult to find, either in text, figures or figure legends. Please provide more info.

Author Response

Reviewer 2

We would like to thank the reviewer for its constructive criticisms and suggestions, and we are confident that we have now addressed to the majority of its concerns.

Major comments:

  • on all figures, labelling of the x axis is a bit confusing, some bars labelled with H2O2, but H2O2 also appears as an adding component for other bars. In figure 1 for example, concentration is indicated for 2 bars but not for the others. In figure 2, H2O2 line does not reach all fractions. Please modify figure presentation for clarify.

Response 1: We have modified the presentations of figures by using signs + and – and also a color code in order to clarify the figures.

  • The muscle cells were used only as myoblasts, and the authors never studies differentiated myotubes (while differentiation is included in Mat&Meths). Proliferating myoblasts represent only a tiny proportion in muscle tissue, and exploration of differentiated myotubes could be of major interest. Please discuss the experimental choice.

Response 2 :  In the first version of the manuscript, we made a mistake by mentioning the differentiation of muscular cells in ‘materials and methods’ section.

We agree with the reviewer that in muscle tissue myoblasts represent only a tiny proportion of cells. However myoblasts are very important for the regeneration of muscle and many studies have reported that the failure of cell therapy in initial clinical trials was mainly attributed to poor survival rates of transplanted myoblasts. This would be due to the fact that myoblasts are very sensitive to oxidative stress compared to differentiated myotubes as reported in our previous study on taxodione (Morel et al 2019, the abietane diterpene taxodione contributes to the antioxidant activity by-product in muscle tissue, J.Funct.Food,62, 2019) and in papers by Salucci et al 2013 (The peculiar apoptotic behavior of skeletal muscle cells. Histology and Histopathology, 28 (8),1073-1087) and Gabor et al 2015 (Differentiation-Associated Downregulation of Poly(ADP-Ribose) Polymerase-1 Expression in Myoblasts Serves to Increase Their Resistance to Oxidative Stress,PloS one, 2015, Vol.10 (7)). In view of these potential clinical implications, we decided to focus all of our experiments to studying the effects of our molecule on the viability of myoblasts.

  • On the human material, all experiments were performed on cells from a single donor. As individual specificity cannot be excluded, limitation of the study should be discussed.

Response 3: it is true that we made all experiments on cells from a single healthy donor. These data should be extended to others clones of different sexes. However, It's far beyond the scope of this exploratory work. To take into account the reviewer comment, we added in the discussion chapter as a limitation  that “the antioxidant efficacy of onopordopicrin should be evaluated in vivo on animal models of skeletal muscle degenerative diseases and ex vivo on human cells derived from patients with myopathies which present an exacerbated susceptibility to oxidative stress”.

  • The number of replicates for each experiment is difficult to find, either in text, figures or figure legends. Please provide more info.

Response 4: we added the number of independent experiments that have been done in the legends of the figures.

Reviewer 3 Report

The authors present a study on bioassay-guided purification of a sesquiterpene lactone onopordopicrin exhibiting antioxidant activity in A. lappa leaf extract. The authors show/suggest that onopordopicrin could be a validl therapeutic molecule for oxidative stress in specific-muscle disorders

Abstract: abbreviations (e.g. HO-1, Nrf) must be explained

Introduction is too general, not specific at all, it should be rewritten to sound more scientific, with explicit details on the topic, active concentration, cell lines, time points for discussed activities, etc., etc., the authors should definitely significantly elaborate on that.

Results: In the sentence on lines 136-139 "These molecules have been 136
reported to possess a broad spectrum of biological activities including anti-inflammatory, phytotoxic, antibacterial, antiplasmodial, antioxidant and cytotoxic/anticancer properties. [31,33-37]", tho following key references are missing:
Sarco/Endoplasmic Reticulum Calcium ATPase Inhibitors: Beyond Anticancer Perspective, Lucie Peterková, Eva Kmoníčková, Tomáš Ruml*, and Silvie Rimpelová, J. Med. Chem. 2020, 63, 5, 1937–1963
Rimpelová, S.; Jurášek, M.; Peterková, L.; Bejček, J.; Spiwok, V.; Majdl, M.; Jirásko, M.; Buděšínský, M.; Harmatha, J.; Kmoníčková, E.; Drašar, P.; Ruml, T. Beilstein J. Org. Chem. 2019, 15, 1933–1944. doi:10.3762/bjoc.15.189
Tailor-Made Fluorescent Trilobolide To Study Its Biological Relevance, Michal Jurášek, Silvie Rimpelová, Eva Kmoníčková, Pavel Drašar, and Tomáš Ruml, J. Med. Chem. 2014, 57, 19, 7947–79

line 145 - not mortality, but rather viability, it is not the same, especially when measured with assays such as WST, MTT, etc., which determine only cell metabolic activity, from which the cell viability is assumed

line 147 - what do you mean by "significantly toxic", please be explicit/specific, where is the line for "significant" toxicity

In plot in Figure 3, part D, the error bars are missing - how many times was the experiment repeated? Since western blot can be only semiquantified, not fully quantified, experiment replicates are necessary, from one western blot membrane with minimal differences in luminescence intensities, you cannot say much, also without the measurement error, nobody knows if the difference for hydrogen peroxide sample is or is not significant, you wrote two independent experiments, but SEM was not counted

the discussion is quite poor, the results and discussion is rather only a description of results, not much discussion is present, especially the relation of sesquiterpene structure to the activity reported and comparison with other structures in the literature and their activities, what drives this antioxidant potential?

Methods: chapter 3.7 different size of font is used
describe in detail how was the differentiation done, also what kind of dishes were the collagen ones

cell death and ROS measurement description is not sufficient, it must be reported in much greater detail so that it is reproducible

siRNA experiment - again, poor description, how many siRNAs against one target were used (normally 6 various are used), also provide the sequences, detailed conditions of the transfection, etc., the methods as they are now, by the authors are totally unrepeatable, almost no details are provided

conditions for SDS-PAGE and Western blot are missing, the code of antibodies are missing, you must state exactly which antibodies did you use for the experiment

the conclusion is rather a brief repetition of methods used, not a real conclusion, it should be rewritten

Author Response

Reviewer 3

We would like to thank the reviewer for its constructive criticisms and suggestions, and we are confident that we have now addressed to the majority of its concerns.

Abstract: abbreviations (e.g. HO-1, Nrf) must be explained

Response 1: we have explained the abbreviations HO-1 and Nrf2 in the abstract.

Introduction is too general, not specific at all, it should be rewritten to sound more scientific, with explicit details on the topic, active concentration, cell lines, time points for discussed activities, etc., etc., the authors should definitely significantly elaborate on that.

Response 2: We added in the introduction sentences to give more explicit details about cell signaling pathway of the muscle cells impacted by oxidative stress: ‘ Many mechanisms have been proposed to explain muscle degeneration caused by the accumulation of ROS. Among these, loss of calcium homeostasis as a result of alteration of the ion channel membrane proteins , mitochondrial dysfunction, activation of NFkB which results in the expression of inflammatory genes, deregulation of autophagy, a mechanism that allows the removal of  defective proteins and organelles’.

We added also more details about previous studies relating antioxidant activity from burdock leaves: In a previous study on A. lappa leaves, Franco et al showed that ethanol extracts present high antioxidant capacities in vitro assays such as scavenging of free radicals (DPPH assay) and oxygen radical absorbance capacity (ORAC assay). Another work also using DPPH assay pointed out the hydroalcoholic extracts of burdock leaves as antioxidant. In this study, Lou and collaborators showed that the antioxidant capacity was associated with the total phenolic content of extracts. Moreover, they suggested that compounds such as chlorogenic acid, o-hydrobenzoic acid, caffeic acid, p-coumaric acid and rutin present in burdock leaves extracts could be involved in antioxidant activity. More recently, Kim et al  showed that a burdock leaves extract has protective effect on age-related macular degeneration model and they suggested that this effect could be due to antioxidant properties of phenolic and flavonoid content present in the leaves extract. Thus, all these studies have reported the potential antioxidant ability of burdock leaves but no specific compound has been isolated from the extracts.

Results: In the sentence on lines 136-139 "These molecules have been 136
reported to possess a broad spectrum of biological activities including anti-inflammatory, phytotoxic, antibacterial, antiplasmodial, antioxidant and cytotoxic/anticancer properties. [31,33-37]", tho following key references are missing:
Sarco/Endoplasmic Reticulum Calcium ATPase Inhibitors: Beyond Anticancer Perspective, Lucie Peterková, Eva Kmoníčková, Tomáš Ruml*, and Silvie Rimpelová, J. Med. Chem. 2020, 63, 5, 1937–1963
Rimpelová, S.; Jurášek, M.; Peterková, L.; Bejček, J.; Spiwok, V.; Majdl, M.; Jirásko, M.; Buděšínský, M.; Harmatha, J.; Kmoníčková, E.; Drašar, P.; Ruml, T. Beilstein J. Org. Chem. 2019, 15, 1933–1944. doi:10.3762/bjoc.15.189
Tailor-Made Fluorescent Trilobolide To Study Its Biological Relevance, Michal Jurášek, Silvie Rimpelová, Eva Kmoníčková, Pavel Drašar, and Tomáš Ruml, J. Med. Chem. 2014, 57, 19, 7947–79

Response 3: We added the 3 references in the list of references.

line 145 - not mortality, but rather viability, it is not the same, especially when measured with assays such as WST, MTT, etc., which determine only cell metabolic activity, from which the cell viability is assumed

Response 4: we have replaced mortality by viability.

line 147 - what do you mean by "significantly toxic", please be explicit/specific, where is the line for "significant" toxicity

Response 5: We removed ‘significantly’and wrote ‘we observed that onopordopicrin was toxic starting from 1 µg/mL with a decrease in the cell viability and an increase in ROS quantity ‘.

In plot in Figure 3, part D, the error bars are missing - how many times was the experiment repeated? Since western blot can be only semiquantified, not fully quantified, experiment replicates are necessary, from one western blot membrane with minimal differences in luminescence intensities, you cannot say much, also without the measurement error, nobody knows if the difference for hydrogen peroxide sample is or is not significant, you wrote two independent experiments, but SEM was not counted

Response 6: We have performed more Western Blot and we made a new figure 3D with error bars from three independent experiments.

the discussion is quite poor, the results and discussion is rather only a description of results, not much discussion is present, especially the relation of sesquiterpene structure to the activity reported and comparison with other structures in the literature and their activities, what drives this antioxidant potential?

Response 7: We changed the text and wrote the results and discussion in different chapters. In addition we wrote in the discussion part a paragraph to propose a structure-activity relationship for onopordopicrin  and we made comparison with other sesquiterpene lactones.

Methods: chapter 3.7 different size of font is used
describe in detail how was the differentiation done, also what kind of dishes were the collagen ones

Response 8:

 We changed the size of font.

Regarding the differentiation, we made a mistake by mentioning the differentiation of muscular cells in ‘materials and methods’ section. There are no results from differentiated cells in the paper. In this study we decided to focus on myoblasts because are very important for the regeneration of muscle and many studies have reported that the failure of cell therapy in initial clinical trials was mainly attributed to poor survival rates of transplanted myoblasts. This would be due to the fact that myoblasts are very sensitive to oxidative stress compared to differentiated myotubes as reported in our previous study on taxodione (Morel et al 2019, the abietane diterpene taxodione contributes to the antioxidant activity by-product in muscle tissue, J.Funct.Food,62, 2019) and in papers by Salucci et al 2013 (The peculiar apoptotic behavior of skeletal muscle cells. Histology and Histopathology, 28 (8),1073-1087)and Gabor et al 2015 (Differentiation-Associated Downregulation of Poly(ADP-Ribose) Polymerase-1 Expression in Myoblasts Serves to Increase Their Resistance to Oxidative Stress,PloS one, 2015, Vol.10 (7)).

The dishes with collagen are: collagen 1 (rat tail tendon)-coated dishes (Corning). We have mentioned this in the section ‘Primary cultures of human myoblasts’ in Materials and Methods.

cell death and ROS measurement description is not sufficient, it must be reported in much greater detail so that it is reproducible

Response 9: We added more details in the section ‘Cell death and ROS quantification’ in Materials and Methods’.

siRNA experiment - again, poor description, how many siRNAs against one target were used (normally 6 various are used), also provide the sequences, detailed conditions of the transfection, etc., the methods as they are now, by the authors are totally unrepeatable, almost no details are provided.

Response 10: We purchased our siRNA At Thermo Fisher Scientific. These siRNA are Silencer Select Pre-designed siRNA, with enhanced specificity to reduce off-target effects.  They guarantee their siRNAs will silence the target mRNA by 70% or more. We added more detail in the section siRNA transfections in ‘Materials and Methods’. We show the results of Western Blot with two siRNAs (siNrf2-91 and siNrf2-93) in Figure 4B and we show also the quantification of this Western Blot indicating that siNrf2-91 reduces 80% of Nrf2 protein level and 60% of HO-1 protein level after normalization with alpha tubulin, our loading control (Figure 4B).

conditions for SDS-PAGE and Western blot are missing, the code of antibodies are missing, you must state exactly which antibodies did you use for the experiment

Response 11: We added in the section ‘Western blotting’ in ‘Materials and Methods’ conditions for SDS-PAGE and Western Blot. We also added the codes of antibodies.

the conclusion is rather a brief repetition of methods used, not a real conclusion, it should be rewritten

Response 12: As we changed the text and wrote the results and discussion in different chapters, we made the choice to have no conclusion part.

Round 2

Reviewer 1 Report

The Reviewer thanks the authors for adressing all important points in an adequate way. The "n" of the performed experiments is still low, however the results now appears consistent and worth it to further elucidate this topic, especially in in vivo settings. 

Author Response

The authors thank the reviewer for his comments

Reviewer 2 Report

Thank you very much for answering the points raised in the first version. I have no further comments.

Author Response

(The authors gave the same response as above.)

Reviewer 3 Report

The authors have answered questions raised and improved the quality of the manuscript.

There are still some minor things to correct, mostly of a formal character, such as e.g. line 100: there have to be a space between a value and a unit (check the whole manuscript in this regard)
The X-axis of Figure 1 - instead of micro (I guess), there is an empty box
"burdock" at some places in the manuscript it is written in Latin, in others, it is not, you should unify it
In Figure 3, there are also some empty boxes instead of signs, the same in some parts of Figure 4
Also, I do not understand, why the conclusion was deleted and now is fully missing?
the methods should be still described in better detail so that they are fully reproducible

Author Response

The authors have answered questions raised and improved the quality of the manuscript.

There are still some minor things to correct, mostly of a formal character, such as e.g. line 100: there have to be a space between a value and a unit (check the whole manuscript in this regard)

Reponse 1: we checked and corrected the ‘minor things’ such as missing spaces between a value and  a unit in the whole manuscript.

The X-axis of Figure 1 - instead of micro (I guess), there is an empty box

Response 2: We corrected the X-axis of Figure 1.

"burdock" at some places in the manuscript it is written in Latin, in others, it is not, you should unify it

Response 3: We chose to use the name in Latin in the whole text.

In Figure 3, there are also some empty boxes instead of signs, the same in some parts of Figure 4

Response 4: In our figure files, we found no empty boxes instead signs in the last versions (sent with the revised manuscript on February 12th) of the Figure 3 and Figure 4 using our PC or MAC computers.

Also, I do not understand, why the conclusion was deleted and now is fully missing?

Response 5: We added a conclusion after the discussion part.

the methods should be still described in better detail so that they are fully reproducible

Response 6: We added more details in the section ‘Primary cultures of human myoblasts’ in Materiels and Methods’. We added also the references of the Muse® Count and Viability Kit and the Muse® Oxidative Stress Kit in the section ‘Cell death and ROS quantification’.